# Adjuvant Docetaxel in Node-Negative Breast Cancer Patients: A Randomized Trial of AGO-Breast Study Group, German Breast Group, and EORTC-Pathobiology Group

**DOI:** 10.3390/cancers15051580

**Published:** 2023-03-03

**Authors:** Christoph Thomssen, Martina Vetter, Eva J. Kantelhardt, Christoph Meisner, Marcus Schmidt, Pierre M. Martin, Florian Clatot, Doris Augustin, Volker Hanf, Daniela Paepke, Wolfgang Meinerz, Gerald Hoffmann, Wolfgang Wiest, Fred C. G. J. Sweep, Manfred Schmitt, Fritz Jänicke, Sibylle Loibl, Gunter von Minckwitz, Nadia Harbeck

**Affiliations:** 1Department of Gynaecology, Martin Luther University Halle-Wittenberg, D-06120 Halle (Saale), Germany; 2Global Health Working Group, Institute of Medical Epidemiology, Biometrics and Informatics, Martin Luther University Halle-Wittenberg, D-06097 Halle (Saale), Germany; 3Institute for Clinical Epidemiology and Applied Biometry, D-72076 Tuebingen, Germany; 4Robert Bosch Society for Medical Research, D-70376 Stuttgart, Germany; 5Department of Gynaecology, Johannes-Gutenberg University, D-55131 Mainz, Germany; 6Department of Medical Oncology, Medical Faculty, F-13344 Marseille, France; 7Department of Medical Oncology, Henri Becquerel Center, F-76038 Rouen, France; 8Department of Gynaecology, Klinikum Deggendorf, D-94469 Deggendorf, Germany; 9Department of Gynaecology, Nathanstift, Hospital Fürth, D-90766 Fürth, Germany; 10Department of Gynaecology, Technische Universitaet Muenchen, D-81675 Munich, Germany; 11Department of Gynaecology, St. Vincenz Hospital, D-33098 Paderborn, Germany; 12Department of Gynecology, St. Josephs-Hospital, D-65189 Wiesbaden, Germany; 13Department of Gynaecology, Katholisches Klinikum, D-55131 Mainz, Germany; 14Department of Laboratory Medicine, Radboud University Medical Center, NL-6500 HB Nijmegen, The Netherlands; 15Department of Gynaecology, University Medical Center Hamburg-Eppendorf, D-20251 Hamburg, Germany; 16German Breast Group Forschungs-GmbH, D-63263 Neu-Isenburg, Germany; 17Breast Center, Ludwig-Maximilian University Hospital, D-81377 Munich, Germany

**Keywords:** node-negative breast cancer, uPA/PAI-1, adjuvant chemotherapy, docetaxel

## Abstract

**Simple Summary:**

The NNBC-3 Europe trial was designed to study the risk assessment in node-negative breast cancer and to test whether taxanes as substitute in an adjuvant anthracycline-containing combination chemotherapy may improve disease-free survival in patients with a high risk of recurrence. We assessed 4146 node-negative breast cancer patients by clinico-pathological or tumor-biological prognostic factors (uPA/PAI-1), and 2541 of these were classified as high-risk and, therefore, treated with six courses of a standard anthracycline combination (FEC) or, by randomization, three courses of the same combination followed by three courses of docetaxel. After a median follow-up of 45 months, we observed only few recurrences without difference between the chemotherapy regimens. With docetaxel, more toxicity was observed. In conclusion, patients with high-risk node-negative breast cancer have an excellent prognosis in the first years after diagnosis independent from the type of chemotherapy. To date, we did not observe sufficient events to evaluate the type of prognostic assessment.

**Abstract:**

Background: In node-negative breast cancer (NNBC), a high risk of recurrence is determined by clinico-pathological or tumor-biological assessment. Taxanes may improve adjuvant chemotherapy. Methods: NNBC 3-Europe, the first randomized phase-3 trial in node-negative breast cancer (BC) with tumor-biological risk assessment, recruited 4146 node-negative breast cancer patients from 2002 to 2009 in 153 centers. Risk assessment was performed by clinico-pathological factors (43%) or biomarkers (uPA/PAI-1, urokinase-type plasminogen activator/its inhibitor PAI-1). High-risk patients received six courses 5-fluorouracil (500 mg/m^2^), epirubicin (100 mg/m^2^), cyclophosphamide (500 mg/m^2^) (FEC), or three courses FEC followed by three courses docetaxel 100 mg/m^2^ (FEC-Doc). Primary endpoint was disease-free survival (DFS). Results: In the intent-to-treat population, 1286 patients had received FEC-Doc, and 1255 received FEC. Median follow-up was 45 months. Tumor characteristics were equally distributed; 90.6% of tested tumors had high uPA/PAI-1-concentrations. Planned courses were given in 84.4% (FEC-Doc) and 91.5% (FEC). Five-year-DFS was 93.2% (95% C.I. 91.1–94.8) with FEC-Doc and 93.7% (91.7–95.3) with FEC. Five-year-overall survival was 97.0% (95.4–98.0) for FEC-Doc and 96.6% % (94.9–97.8) for FEC. Conclusions: With adequate adjuvant chemotherapy, even high-risk node-negative breast cancer patients have an excellent prognosis. Docetaxel did not further reduce the rate of early recurrences and led to significantly more treatment discontinuations.

## 1. Introduction

Today, taxanes are part of adjuvant standard chemotherapy in high-risk breast cancer. Particularly in node-negative breast cancer, absolute improvement of survival is small [1,2], and higher benefit can only be expected in patients at high-risk of recurrence. Risk assessment in node-negative breast cancer is routinely performed by using established clinico-pathological algorithms [3]. However, since 2007, ASCO guidelines as well as German AGO guidelines have also recommended the invasion markers urokinase-type plasminogen activator (uPA) and its inhibitor PAI-1 for risk assessment and treatment decision [4,5,6] (https://www.ago-online.de/leitlinien-empfehlungen/leitlinien-empfehlungen/kommission-mamma) (accessed on 28 February 2023).

More than 25 years ago, Jänicke [7,8,9] demonstrated that the risk of recurrence in node-negative breast cancer can be more effectively characterized by uPA/PAI-1 than by conventional markers: with low uPA/PAI-1, without any adjuvant systemic therapy, a remarkably low risk of recurrence was seen in more than half of all node-negative patients. In the prospective Chemo-N0-trial [10,11,12], we have validated the independent and strong prognostic impact of uPA/PAI-1 with regard to a disease-free survival (DFS) and an overall survival (OS) and demonstrated that particularly patients with high uPA/PAI-1 substantially benefit from adjuvant CMF chemotherapy. The prognostic value and the particular benefit from adjuvant chemotherapy associated with high uPA/PAI-1 was also confirmed in large pooled analyses [13,14,15].

In multivariate analysis of DFS, only grade, uPA/PAI-1 and young age were strong and independent prognostic markers [10,12]. In a subgroup analysis, uPA/PAI-1 differentiate between a low and a high risk of recurrence also in patients with intermediate histopathological grade (G2). Based on this prospective data, tumor-biological risk assessment using invasion markers uPA/PAI-1 was proposed to challenge conventional clinico-pathological risk assessment [3,16,17].

Assessing the actual risk of recurrence effectively, adjuvant chemotherapy may be spared in a majority of node-negative patients. Particularly in patients with high uPA/PAI-1, efficacy of adjuvant chemotherapy is improved; this benefit may be enhanced by integrating taxanes into conventional anthracycline-containing therapy [2]. Thus, the NNBC 3-Europe trial (AGO-B-011) had two major questions:To investigate whether substituting the last three courses of a standard adjuvant FE_100_C (six courses) by three courses of docetaxel improves the disease-free survival of high-risk node-negative breast cancer patients.To quantify the effects of the tumor-biological risk assessment (uPA/PAI-1) and the clinico-pathological risk assessment with regard to disease-free survival and the proportion of low-risk patients in node-negative breast cancer patients.With current follow-up, data are only mature to report on the first question.

## 2. Materials and Methods

### 2.1. The Objectives

Primary objective of the NNBC 3-Europe chemotherapy part was to demonstrate a clinically relevant improvement of disease-free survival (DFS) by using FEC-Doc instead of FEC as adjuvant chemotherapy in high-risk node-negative breast cancer. Loco-regional recurrences, distant metastases, and death were considered as DFS events. Secondary objectives were a description of OS and toxicity.

### 2.2. Patient Population

Female patients between 18 and 70 years of age were eligible if they had histologically confirmed node-negative breast cancer with a tumor size between 0.5 cm and 5 cm. Axillary lymph-node evaluation was performed by at least 10 dissected nodes or by a sentinel procedure. For centers that used tumor-biological risk-assessment, frozen tissue had to be available for uPA/PAI-1 testing.

### 2.3. Study Design and Procedures

NNBC 3-Europe is a multicenter, prospective, randomized, not blinded, controlled trial. After patients were registered for the trial, risk-assessment was performed either by tumor-biological or by clinico-pathological means (Figure 1). Patients at high risk of recurrence were randomized using a 1:1 permuted-block randomization via the internet, stratified for center, the type of risk-assessment, and HER2-status to receive three courses 5-fluorouracil 500 mg/m^2^, epirubicin 100 mg/m^2^, and cyclophosphamide 500 mg/m^2^ (FEC) followed by three courses of docetaxel 100 mg/m^2^ (Doc) or six courses of standard FEC [18]. Therapy had to be started within 6 weeks after axillary dissection. Patients with a low risk of recurrence received standard adjuvant endocrine treatment alone. All patients had to have adequate loco-regional therapy (mastectomy or breast conserving therapy) according to the individual decision by the treating physician. Radiotherapy, endocrine therapy, and trastuzumab were recommended according to national guidelines at the time (www.ago-online.org, accessed on 28 February 2023 [5]).

### 2.4. Mode of Risk Assessment

Risk of recurrence was assessed either by tumor-biological or by clinico-pathological criteria, and patients were then classified as low or high risk. Histopathological evaluation was performed by local pathology institutions. Stage, morphological, and immune-histochemical results were used as reported for clinical routine use.

For tumor-biological risk assessment, frozen tissue of the primary tumor was shipped to one of the designated laboratories for uPA and PAI-1 determination. Taking into account the results of the CHEMO-N0 trial [12], patients were considered at high risk, if they were =<35 years or they had grade 3 (G3) tumors or if tumors were, independent from age, grade 2 (G2) with high uPA/PAI-1 tumor concentrations. All others were defined as low risk (Appendix A).

For clinico-pathological risk assessment, as published previously [3], patients were assigned to the high-risk group if at least one of the following characteristics was given: age <40 yrs, G3, HER2-positive, PR-negative, vessel invasion, and G2-tumors if ≥2 cm. (Appendix A).

Since randomization of the type of risk assessment did not seem feasible, each of the 153 study centers chose its method of their individual risk assessment upfront. Fifty-six centers chose the tumor-biology-based risk assessment, and 97 centers chose the clinico-pathological risk assessment. In order to reduce a potential bias by the quality of a center, all centers should have at least internet access, which was not standard at the time, and the facility to store deep frozen samples. In addition, the centers should in general perform their treatment decisions according to the national AGO recommendations [www.ago-online.de], which were required.

### 2.5. uPA and PAI-1 Assessment

Of each surgically excised tumor, a 300–500 mg specimen of fresh tissue was collected by the responsible pathologist of each center, immediately snap frozen in liquid nitrogen, and sent on dry ice to one of the uPA/PAI-1-trial labs (listed in Appendix A). The tissue was pulverized, and protein extraction was performed over night at 4 °C in 300–1000 µL of Tris extraction buffer (Tris-buffered saline: 50 mM Tris pH 8.5, 138 mM NaCl, 2.7 mM KCl with 1% (*w*/*w*) Triton X-100). After centrifugation at 13,000× *g* for 1 h at 4 °C, the supernatants were used for analysis.

Quantification of uPA and PAI-1 antigen levels was performed by ELISA using the certified FEMTELLE™ Kit (American Diagnostica Inc., Stamford, CT, USA, now LOXO GmbH, Dossenheim, Germany). Total protein concentration was determined using the Pierce^®^ BCA protein assay (Pierce, Rockford, IL, USA). Concentrations of uPA and PAI-1, respectively, were set in relation to the total protein concentration of the tumor lysate and reported in ng analyte per mg total protein.

### 2.6. Quality Assurance of uPA and PAI-1 Testing

Fourteen participating laboratories in Germany (n = 12) and France (n = 2) used the same tissue extraction and determination methods, including the calibrator for total protein determination to achieve a maximum degree of standardization [19,20,21]. Internal and external quality assurance (QA) concerning within-assay, within-laboratory, between-assays, and between-laboratory quality, respectively, was implemented. In total, corresponding samples with 18 vials for uPA and PAI-1-testing and 18 vials for total protein-testing were used in annually repeated inter-laboratory ring trials (Appendix A).

### 2.7. Data Collection and Statistical Analysis

Randomization and data collection were performed on a web-based platform (Trium Analysis Online GmbH, Munich, Germany). Monitoring was performed by GBG (GBG Forschungs-GmbH, Neu-Isenburg, Germany). For final analysis, data were transferred as a cleaned SAS database to the Institute for Clinical Epidemiology and Applied Biometry, Tuebingen, Germany. Statistical analysis was performed with SAS 9.2.

The sample size for DFS was computed assuming an accrual period of three years and a follow-up time of 2.5 years, resulting in a maximum follow-up time of 5.5 years for the first randomized patient. Assuming a recurrence rate of 13% at 5 years of follow-up in the standard FEC arm, an absolute difference of 4% was considered as clinically relevant (i.e., an anticipated recurrence rate of ≤9% in the FEC-Doc arm) allowing a drop-out of 7% after 2.5 years. Under these assumptions, a sample size of 1286 for each study arm was determined for a two-sided log rank test with significance level of 0.050 and 80% power.

Statistical analyses were performed according to an analysis plan that was established prior to the start of the trial and adapted according to the course of recruitment. Missing values were depicted in Table 1. Patients with missing data in one or more prognostic factors were excluded from survival analysis. For each patient, the primary endpoint, disease-free survival (DFS), was calculated in days as time elapsed between surgery and first documented DFS-event defined as any recurrence, such as distant metastasis, loco-regional relapse, or breast cancer related death corresponding to RFI according to Hudis et al., 2007 [22]. For statistical evaluation, event-free patients were censored at the date of last follow-up. Patients who died for reasons not related to breast cancer were censored at the date of death. Prespecified secondary endpoints were overall survival (OS) and acute toxicity from chemotherapy.

Univariate descriptive analyses of DFS and OS were performed with Kaplan–Meier estimates and Cox’s proportional hazards models to estimate hazard ratios (HR) with two-sided 95% confidence intervals (C.I.). To test for superiority of one therapy, two-sided log rank tests were used. Multivariate analyses of DFS and OS were performed using Cox’s proportional hazards models to adjust for prespecified confounding factors: age (>50 vs. ≤50), tumor size (pT2 vs. pT1), grade (G3 vs. G2), type of loco-regional treatment (BCT vs. MRM), estrogen receptor status (negative vs. positive), progesterone receptor status (negative vs. positive), HER2-status (positive vs. negative), vessel invasion (present vs. not-present), type of risk group assessment (clinical-pathological vs. tumor-biological), and number of chemotherapy courses delivered (<6 courses vs. complete). To assess for independent prognostic factors of DFS, multivariate analyses were applied. Toxicity was reported in a descriptive manner using the NCI Common Toxicity Criteria Version 2.0 scale. Analyses of the primary and secondary efficacy endpoints were based on all randomized patients (i.e., intention-to-treat, ITT, population). Patients without any dosage of chemotherapy were excluded from ITT. The per-protocol analyses of the primary endpoint included only patients without major protocol violations (i.e., per-protocol population, PP; details in the CONSORT diagram, Figure 2).

Finishing the planned statistical analyses, we decided to include additional explorative analyses concerning the therapeutic effect in some subgroups. For these analyses, HR with two-sided 95% C.I. derived from Cox’s proportional hazards models of DFS were estimated and presented in a forest plot.

The participating centers are listed in the Appendix A. Patients were registered only if they had given written informed consent. The trial was conducted in accordance with the Declaration of Helsinki, and it is registered with Clinicaltrials.gov (NCT01222052). The protocol was reviewed and cleared by all responsible local ethics committees. None of the supporting companies had any role in the study design, data collection, data analysis, data interpretation, or writing of the report. Safety was analyzed nine times during the study and discussed with an independent data and safety monitoring board.

## 3. Results

### 3.1. Patient Population

Between 2002 and 2009, 4146 node-negative breast cancer patients were registered, 1481 patients were evaluated to be at low risk of recurrence and not included into the randomized part of the trial, and five patients were excluded as registration failure. The 2660 patients at a high risk of recurrence were randomized to receive either FEC-Doc (n = 1333) or FEC (n = 1327) as adjuvant chemotherapy. In total, 119 patients did not receive any chemotherapy and were excluded from ITT analysis. Consequently, the ITT population consists of 2541 randomized patients (FEC-Doc: 1286; FEC: 1255). Tumor-biological risk assessment had been performed in 1452 patients (57.1%) and clinico-pathological risk assessment in 1089 (42.9%). In July 2012, 90.8% of patients had reached the minimum observation time of 30 months (89.0% treated by FEC-Doc and 92.6% treated by FEC) with a median follow-up of 45 months. For a confirmatory per-protocol analysis, a further 244 FEC-Doc arm and 140 FEC arm patients needed to be excluded, mostly because of non-adequately delivered chemotherapy (i.e., <6 chemotherapy courses, Figure 2).

### 3.2. Patient Characteristics

The main patient characteristics were equally distributed between chemotherapy arms (Table 1): median age was 53 yrs (21–70), 41.4% of patients were peri- or premenopausal, median tumor size was 1.9 cm, grade 3 was seen in 53.8%, negative steroid hormone receptor status in 30.1% and positive HER2-status in 20.0%, and mastectomy was performed in 11% of all tumors. Complete chemotherapy delivery was obtained for FEC-Doc in 84.4% and for FEC in 91.5% (*p* < 0.0001).

### 3.3. Disease-Free Survival

DFS of the patients was excellent in both groups: 66 patients experienced a DFS event in the FEC-Doc arm (30 patients within the first 2.5 years) and 61 in the FEC arm (33 patients within the first 2.5 years). The Kaplan–Meier estimates for 5 years disease-free survival was 93.2% (95% C.I. 91.1–94.8) in the FEC-Doc arm and 93.7% (95% C.I. 91.7–95.3) in the FEC arm (*p* = 0.807) with an unadjusted HR of 1.05 (95% CI 0.74–1.48) (Figure 3A). In the multivariate analysis, an adjusted HR of 0.98 (95% CI 0.68–1.40) was estimated. The results were confirmed by the PP analysis (unadjusted HR = 1.01 (95% C.I. 0.69–1.50), adjusted HR = 1.04 (95% C.I. 0.70 –1.54); *p* = 0.948). Analysis of DFS in different subgroups confirmed the main analysis with no subgroup deriving particular benefit from either chemotherapy. (Figure 4).

### 3.4. Overall Survival

Overall, we observed 55 breast cancer related deaths, 27 in the FEC-Doc arm and 28 in the FEC arm. The Kaplan–Meier-estimates for five-year overall survival was 97.0% (95% C.I., 95.4–98.0, FEC-Doc) and 96.6% (95% C.I. 94.9–97.8, FEC), respectively, with an adjusted HR of 0.98 (95% C.I. 0.68–1.40). The results were confirmed in the PP population (adjusted HR = 0.81 (95% C.I. 0.48–1.41) (Figure 3B).

### 3.5. Prognostic Factors

In a multivariate analysis of DFS, tumor size (>1 cm; HR 2.94, 95%-C.I. 1.08–8.0, *p* = 0.035), incomplete chemotherapy (<6 courses; HR 2.06, 95% C.I. 1.25–3.38, *p* = 0.005), negative PR-status (HR 1.96, 95% C.I. 1.17–3.29, *p* = 0.01), and negative ER status (HR 1.89, 95% C.I. 1.13–3.165, *p* = 0.015) were significant and independent prognostic factors (Table 2). In a multivariate analysis of overall survival, only incomplete chemotherapy (HR 3.92, 95% C.I. 2.10–7.31, *p* < 0.0001) and age >50 (HR 1.88, 95% C.I. 1.03 3.46, *p* = 0.041) had a significant independent prognostic impact. We performed a separate exploratory analysis for patients with luminal breast cancer; however, similarly no additional benefit by integrating docetaxel has been demonstrated (Appendix A).

### 3.6. Toxicity

Both chemotherapies, FEC and FEC-Doc, are well described and broadly used. Nevertheless, adverse events were reported for each of the 14,538 chemotherapy courses according to the CTCAE-Scale (v2.0). For the 7245 courses in the FEC-arm, overall, 9113 events (1430 grade 3/4) in 969 patients (461 grade 3/4) were reported, for the 7293 courses in the FEC-Doc-arm, and 9873 events (1708 grade 3/4) in 1030 patients (562 grade 3/4). One possible therapy-related death was reported in a patient who received FEC-Doc and experienced a severe sepsis with encephalitis immediately after the first course of docetaxel. The autopsy did not show a distinct entry focus. Overall, patients treated by FEC showed better tolerance of the chemotherapy (all grades *p* = 0.081), for grade 3/4 events the difference was significant (*p* < 0.0004). More detailed information is depicted in Appendix A.

## 4. Discussion

Today, with modern therapy concepts, patients with early breast cancer have an excellent prognosis [23]. We showed that this is particularly true for high-risk node-negative breast cancer patients who received standard adjuvant chemotherapy. Partial substitution of FEC-courses by docetaxel had been associated with a significant improvement of DFS in other trials [24,25,26]. Yet, in our trial that focused on high-risk node-negative breast cancer patients defined by either clinico-pathological or tumor-biological criteria, it had no additional beneficial effect over an anthracycline-containing combination regimen and even led to significantly more treatment discontinuations. In summary, our results fit well with the observation of the Oxford overview that taxanes only reduce the rate of recurrence if given in addition to standard anthracycline-containing therapy, but not if they merely replace some non-taxane chemotherapy courses [25].

It has to be mentioned that another trial evaluating the effect of docetaxel in node-negative breast cancer had a similarly excellent outcome and did not show a difference between docetaxel-containing and standard therapy at 30 months follow-up but showed a significant benefit on DFS with longer follow-up [2]. In addition, also in the PACS01-trial that had an identical design as NNBC3-Europe but studied node-positive patients, a relevant effect of docetaxel was observed only with longer follow-up [24,27].

The NNBC 3-Europe study population is representative for a high-risk node-negative breast cancer cohort also according to conventional factors: more than half of the patients had undifferentiated tumors, more than a third were premenopausal, 30.1% had ER- and PR-negative tumors, and 20.0% HER2-positive tumors. Thus, it is rather unlikely that an inadvertent selection of low-risk patients would have influenced the results. Moreover, in an exploratory analysis, there was no subgroup of patients for whom a particular benefit from addition of docetaxel could be demonstrated. Additionally, the type of risk assessment, including uPA/PAI-1, did not predict DFS or OS (Figure 4).

In multivariate analyses, we demonstrated an independent prognostic effect of tumor size, incompletely administered chemotherapy, ER and PR on DFS and of age, and incomplete chemotherapy on OS within this high-risk group of node-negative breast cancer patients (Table 2). Thus, we can conclude that adjuvant chemotherapy was well indicated: patients who stopped the treatment early had a significantly worse DFS and OS than those who completed the planned courses of treatment. However, this result disagrees with data from the PACS05 trial that did not show a difference in efficacy between four and six courses of FEC in a similar setting [28]. We are well aware that today patients with HER2-positive or triple-negative breast cancer would be treated differently [29,30]; therefore, we performed a separate exploratory analysis for patients with luminal breast cancer where we did not find different results (Appendix A). It could be hypothesized that FEC-Doc was not superior to FEC because it created additional toxicity. With regard to grade 3 and 4 events alone, we observed more infections, arthralgia, diarrhea, neuropathy, and allergic reactions with FEC-Doc. Overall, more patients discontinued chemotherapy early in the FEC-Doc (15.6%) than in the FEC arm (8.5%), and early discontinuation was one of the strongest independent prognostic factors in our study. Considering that also the addition of 5-fluorouracil does not improve the chemotherapy effect [31], today, less toxic sequential regimens have become standard (e.g., dose-dense EC followed by weekly paclitaxel).

Another explanation for the lack of an additional effect of docetaxel may be drawn from the Oxford overview (EBCTCG, [25]) that suggested the benefit of a taxane addition, particularly in higher differentiated (G1 and G2), but not in undifferentiated, cancers (G3). However, at least one trial (PACS01) with a similar design as ours did not show a differential effect with regard to tumor grading [24,27].

High levels of uPA/PAI-1 are predictive for an enhanced benefit from adjuvant chemotherapy [13]. However, despite treating with effective therapies, we still observe some recurrences. Incompletely delivered chemotherapy has been demonstrated to be the most important predictor; however, multiple other reasons can be discussed, including primary resistance to chemotherapy, failing diagnosis of metastatic disease, or suboptimal design or lack of adherence for endocrine therapy. In order to further reduce the risk of recurrence, today, more effective therapies, including targeted therapies, would be administered.

A possible weakness of our study is that enrollment was slower than expected. This could be due to the fact that other risk assessment tools, such as gene expression signatures, are increasingly used. Another possible limitation is that the predefined short follow-up period only allowed the assessment of the five-year survival rate. However, with more than 30% hormone-receptor-negative patients, it is unlikely that a significant effect on DFS was missed even in this short follow-up period. A strength of our study is that we were able to show that the replacement of anthracyclines with docetaxel did not lead to an improved survival rate but to increased toxicity in a large and well-defined cohort of node-negative breast cancer patients with a high risk of recurrence.

With regard to the second question on the prognostic impact of uPA/PAI-1 in comparison to clinico-pathological criteria, we have not reached the number of necessary events so far. Preliminary observation suggests that the patients in the low-risk group may have a very low number of recurrences, confirming that considering them as low risk was correct by both types of risk assessment. Nevertheless, formal analysis still needs to be performed.

## 5. Conclusions

The prospective randomized multicenter NNBC 3-Europe trial is the first phase III trial in node-negative breast cancer that prospectively compared tumor-biological against clinico-pathological risk assessment for identifying high-risk patients as candidates for adjuvant chemotherapy. Here, we show that with state-of-the art therapy patients with high-risk node-negative breast cancer as defined by one of these methods have an excellent prognosis. In this high-risk subgroup, use of docetaxel instead of three more courses of an anthracycline-containing combination did not further reduce the rate of early recurrences but led to significantly more treatment discontinuations. As an incidental observation, we found that in patients with a high risk for recurrence, delivering chemotherapy in the planned manner by avoiding treatment discontinuations significantly improves outcome.

## Figures and Tables

**Figure 1 cancers-15-01580-f001:**
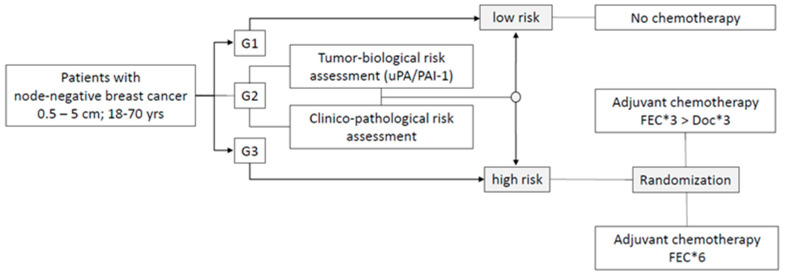
NNBC 3-Europe trial design. All patients with hormone-receptor-positive tumors received endocrine therapy according to local standards (in most cases tamoxifen 20 mg PO/die for 5 years).

**Figure 2 cancers-15-01580-f002:**
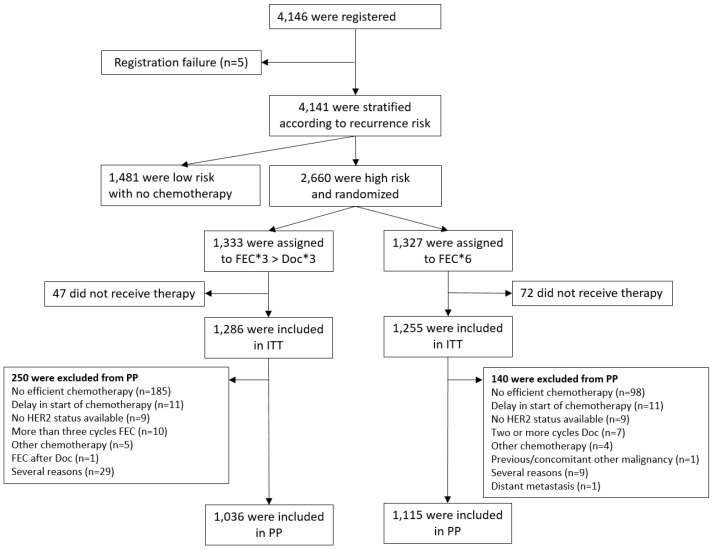
CONSORT Diagram. Only patients at high risk of recurrence, assessed by either tumor-biological or clinico-pathological means (see Appendix A), were included in the randomized part of the trial. (C—Cyclophosphamide, Doc—docetaxel, E—epirubicin, F—5-fluorouracil; HER2—human epidermal growth factor receptor 2; ITT—intention-to-treat cohort, PP—per-protocol cohort). Note: some patients had two (n = 34) or three (n = 4) protocol violations, simultaneously.

**Figure 3 cancers-15-01580-f003:**
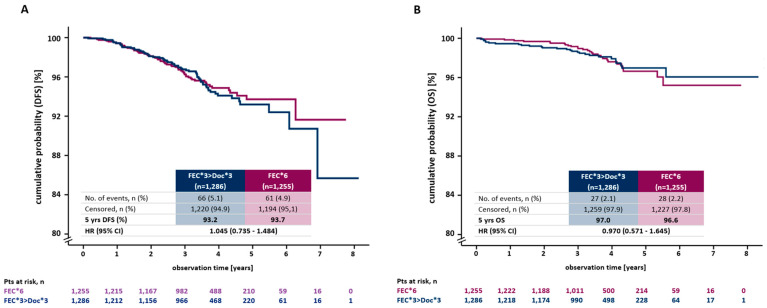
Survival estimates for DFS (**A**) and OS (**B**) stratified by FEC*3 > Doc*3 and FEC*6. The tables present the effective sample size for each interval (numbers at risk).

**Figure 4 cancers-15-01580-f004:**
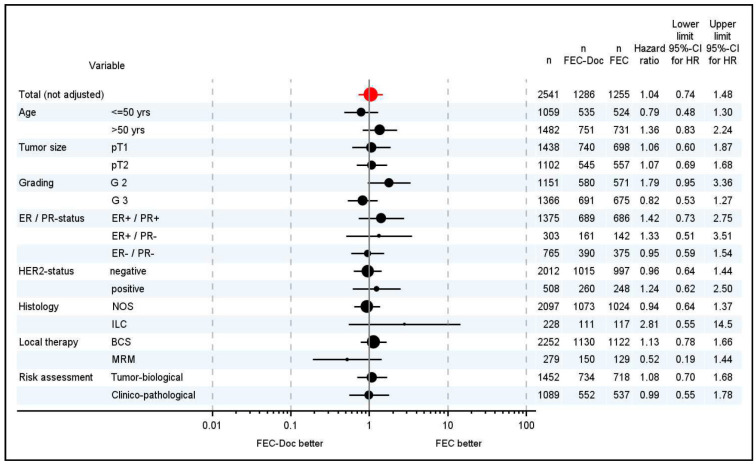
Forest plot. Subgroup analysis for disease-free survival (number of events in Appendix A). Abbreviations: estrogen receptor (ER), progesterone receptor (PR), human epidermal growth factor receptor 2 (HER2), not otherwise specified (NOS), invasive lobular cancer (ILC), breast conserving therapy, including radiotherapy (BCS), modified radical mastectomy (MRM).

**Table 1 cancers-15-01580-t001:** Patients’ and tumors’ characteristics, ITT-cohort n = 2541 pts.

	Total (ITT) n = 2541		FEC*3 > Doc*3 n = 1286		FEC*6 n = 1255	
Variable	n	(%)	n	(%)	n	(%)
Age (mean)	52.6 y		52.7 y		52.5 y	
Peri-/premenopausal	1051	(41.4%)	524	(40.7%)	527	(42.0%)
Postmenopausal	1442	(56.7%)	737	(57.3%)	705	(56.2%)
Missing	48	(1.9%)	25	(1.9%)	23	(1.8%)
Tumor size (median/mean)	1.9 cm/2.98 cm		1.9 cm/3.08 cm		2.0 cm/2.89 cm	
pT1b	222	(8.7%)	130	(10.1%)	92	(7.3%)
pT1c	1216	(47.9%)	610	(47.4%)	606	(48.3%)
pT2	1102	(43.4%)	545	(42.4%)	557	(44.4%)
Missing	1	(0.1%)	1	(0.1%)	-	-
Invasive ductal cancer	2097	(82.5%)	1073	(83.4%)	1024	(81.6%)
Invasive lobular cancer and others	421	(16.6%)	200	(15.6%)	221	(17.6%)
Missing	23	(0.9%)	13	(1.0%)	10	(0.8%)
Grade 1	14	(0.6%)	9	(0.7%)	5	(0.4%)
Grade 2	1151	(45.3%)	580	(45.1%)	571	(45.5%)
Grade 3	1366	(53.8%)	691	(53.7%)	675	(53.8%)
Missing	10	(0.4%)	6	(0.5%)	4	(0.3%)
Hormone receptor status positive (ER-pos. and/or PR-pos.)	1770	(69.7%)	893	(69.4%)	877	(69.9%)
Hormone receptor status negative (ER-neg. and PR-neg.)	765	(30.1%)	390	(30.3%)	375	(29.9%)
Missing	6	(0.2%)	3	(0.2%)	3	(0.2%)
HER2-positive	508	(20.0%)	260	(20.2%)	248	(19.8%)
HER2-negative	2012	(79.2%)	1015	(78.9%)	997	(79.4%)
Missing	21	(0.8%)	11	(0.9%)	10	(0.8%)
Vessel invasion (present)	195	(7.7%)	100	(7.8%)	95	(7.6%)
Vessel invasion (not present)	1718	(67.6%)	867	(67.4%)	851	(67.8%)
Missing	628	(24.7%)	319	(24.8%)	309	(24.6%)
Mastectomy	279	(11.0%)	150	(11.7%)	129	(10.3%)
Breast conserving therapy (BCT)	2252	(88.6%)	1130	(87.9%)	1122	(89.4%)
Missing	10	(0.4%)	6	(0.5%)	4	(0.3%)
Biological (uPA/PAI-1) risk assessment	1452	(57.1%)	734	(57.1%)	718	(57.2%)
Clinico-pathological risk assessment	1089	(42.9%)	552	(42.9%)	537	(42.8%)
Chemotherapy completed *	2234	(87.9%)	1086	(84.4%)	1148	(91.5%)
Chemotherapy incomplete <6 courses	307	(12.1%)	200	(15.6%)	107	(8.5%)
*Chemotherapy incomplete* *≤4 courses*	*196*	*(7.7%)*	*124*	*(9.6%)*	*72*	*(5.7%)*
Follow-up ≥ 2.5 y	2306	(90.8%)	1144	(89.0%)	1162	(92.6%)
Distant recurrences	94	(3.7%)	50	(3.9%)	44	(3.5%)
Loco-regional recurrences	51	(2.0%)	24	(1.9%)	27	(2.2%)
Second cancers	39	(1.5%)	20	(1.6%)	19	(1.5%)
Deaths	55	(2.2%)	27	(2.1%)	28	(2.2%)

* *p*-value (Pearson χ^2^ test) < 0.001. Abbreviations: intention-to-treat (ITT), estrogen receptor status (ER), progesterone receptor status (PR), human epidermal growth factor receptor 2 (HER2), urokinase-type plasminogen activator (uPA), plasminogen activator inhibitor type 1 (PAI-1).

**Table 2 cancers-15-01580-t002:** Univariate and multivariate analysis of disease-free survival and overall survival.

Variable	Univariate Analysis	Multivariate Analysis		
	*p*-Value	Hazard Ratio	95% Confidence Interval	*p*-Value
**Disease-free survival**				
Tumor size	0.03	2.94	1.08–8.0	0.0355
(>1 cm vs. ≤1 cm)				
Chemotherapy	0.001	2.06	1.25–3.38	0.005
(<6 courses vs. complete)				
Estrogen receptor (ER) status	<0.0001	1.89	1.13–3.17	0.015
(negative vs. positive)				
Progesterone receptor (PR) status	<0.0001	1.96	1.17–3.29	0.01
(negative vs. positive)				
Vessel invasion	0.009	1.74	0.99–3.06	0.056
(present vs. not present)				
Grading (G3 vs. G2)	0.005	1.06	0.69–1.61	0.803
Age	0.095	0.76	0.53–1.09	0.133
(>50 years vs. ≤50 years)				

**Overall Survival**				
Chemotherapy	<0.0001	3.92	2.10–7.31	<0.0001
(<6 courses vs. complete)				
Estrogen receptor status	0.038	1.3	0.62–2.72	0.484
(negative vs. positive)				
Progesterone receptor status (negative vs. positive)	0.008	1.89	0.91–3.92	0.086
Age	0.045	1.88	1.03–3.46	0.041
(>50 years vs. ≤50 years)				

ER and PR status considered positive by local pathologist according to the definition at the time.

## Data Availability

The data generated in this study are available within the article and its Appendix A. Raw data were generated and processed from the authors and are available on request to the corresponding authors.

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
