# Peer review of "Adjuvant Docetaxel in Node-Negative Breast Cancer Patients: A Randomized Trial of AGO-Breast Study Group, German Breast Group, and EORTC-Pathobiology Group"

_cancers, 2023, doi:10.3390/cancers15051580_

Round 1

Reviewer 1 Report

Dear colleagues,

It was a great honor to get acquainted with your work, which, in my opinion, turned out to be interesting and will find its reader.

In this randomized, multicenter study, the authors were investigating the use of adjuvant therapy according to the assessed risk of recurrence. Patients with node-negative breast cancer at high risk of recurrence, which was defined using clinic-pathological and tumor-biological characteristics, received either FEC regimen which was the standard of treatment at the time of planning the study, or the FEC-Doc regimen. Disease-free survival was assessed as the primary endpoint.

When prescribing adjuvant therapy, we always want to be more confident that it is necessary for this particular patient due to additional toxicity and the inevitable overtreatment of some patients. Prescription of one or another generation of chemotherapy based on the patient's prognosis and risk factors is a concept that has become quite firmly established in breast cancer routine practice therapy starting from validated risk-calculators to genomic signatures.

The conducted study is understandable and clear and the manuscript as a complete work is generally well written. However, some issues and precise details need a correction.

Specific comments

1.       Abstract and introduction are well written and they indicate the background and hypothesis of the study.

2.       Materials and methods are clear and well defined, except for some minor moments:

a.       p. 3-4 lines 135-136 “patients were considered at high risk <….> if tumors were grade 2 (G2) with high uPA/PAI-1 tumor concentrations.” It will be more clear if age >35 will be specified here.

b.       p. 5 line 201 Loco-regional treatment includes axillary surgery which is not specified.

c.       p. 5 line 211 Definition of “major protocol violations” seems to be needed or a reference to a CONSORT diagram should be made.

3.       CONSORT diagram: “Two or more cycles of FEC”. The meaning is not clear for interpretation. Sounds like 6 cycles also fit in this category and less than two are not.

4.       Results chapter also need some corrections:

a.       p. 6 line 246 maximal age was 70 years and the inclusion criteria are less than 65. How many patients were included despite the inclusion criteria? Were there violations of other inclusion criteria?

b.       p.6 line 246 percent of patients who were premenopausal do not match with percent in Table 1.

c.       p.6 first paragraph. It will be more visually easy to perceive information if there will be a min-max range for tumour size and absolute numbers with percents for other values.

d.       Table 1. HER2 status missing information is not consistent with the information from CONSORT diagram in which unavailable HER2 status was an exclusion criterion from per-protocol analysis.

e.       Table 1. If you define Breast Surgery and Axillary surgery separately it would be appropriate to define Sentinel lymph node biopsy also.

f.        p.8 line 257-258 numbers of patients with DFS events does not meet the numbers in table 1.

g.       p.8 line 259 Seems like you mean disease free survival and not disease free interval.

h.       p. 9 Toxicity paragraph. It is very hard to perceive information about adverse effects in absolute number of cycles performed. Maybe it would be better to present them in percentage of patients in whom particular adverse effect was developed.

5.       First sentence of conclusion states that this is the first phase 3 trial in NNBC which compared different risk assessment systems and that’s true, but the article null hypothesis was not about that statement and that’s why this sentence is irrelevant to the article and results.

Major comments

·         This is well designed study and the idea is clear and understandable. Nevertheless, from the start of the accrual till nowadays more than 20 years past. Nowadays we have modern standards of therapy with excellent oncologic outcomes which exceed numbers achieved in this study. In our daily practice we are using new surgical techniques and new regimens of systemic therapy which are obviously not taken into consideration as the trial was designed over 2 decades ago. It is an interesting work but it’ll be great if you will specify its meaning for routine clinical practice more precisely.

Author Response

Dear Prof. Antonis Valachis,

first, we would like to thank for your evaluation, the critical discussion of our manuscript and the opportunity to resubmit a revised copy of our manuscript. We would also like to take this opportunity to express our thanks to the reviewers for the feedback and helpful comments for corrections and modifications. We have carefully considered the comments of the reviewers and improved the manuscript accordingly. Also, we have enclosed a point-by-point response to each of the reviewers' comments and the changes are made for the resubmission. Comments are in blue and the modified sentences in blue/bold. In addition we are going to send you the marked manuscript.

You can find the improved revised manuscript alongside this e-mail and on the website. We look forward to hearing from you. If you have any further questions, please don’t hesitate to contact us.

With best regards

Christoph Thomssen

Corresponding Author:
Prof. Dr. med. Christoph Thomssen
Martin-Luther-University Halle-Wittenberg
Ernst-Grube-Strasse 40, 06097 Halle (Saale), Germany
Phone: +49 345 557 1366
E-mail: christoph.thomssen@medizin.uni-halle.de

The reviewers' comments are answered point by point and the responses are indicated in blue. The changes made are referenced to page and paragraph from the resubmited manuscript.

The changes are simultaneously applied and indicated in the manuscript's new version according to the Journal´s guidelines.

Reviewer 1

The conducted study is understandable and clear and the manuscript as a complete work is generally well written. However, some issues and precise details need a correction.

[Comment 1]

Abstract and introduction are well written and they indicate the background and hypothesis of the study

[Answer 1]

Thank you. We appreciate your statement.

[Comment 2a]

  1. 3-4 lines 135-136 “patients were considered at high risk <….> if tumors were grade 2 (G2) with high uPA/PAI-1 tumor concentrations.” It will be more clear if age >35 will be specified here.

[Answer 2a]

Logically, in our opinion, the statement is clear: Or high for “=<35 years or they had grade 3 (G3)” or high risk for “grade 2 (G2) with high uPA/PAI-1 tumor concentrations”. However, we have added the wording “independent from age” to make it more clear (line 135, 136)

[Comment 2b]

  1. 5 line 201 Loco-regional treatment includes axillary surgery which is not specified

[Answer 2b]

Axillary surgery is defined in the section 2.2 Patient population: “Axillary lymph-node evaluation was performed by at least 10 dissected nodes or by sentinel procedure.” (line 107+). We would like to consider that as sufficiently described.

[Comment 2c]

  1. 5 line 211 Definition of “major protocol violations” seems to be needed or a reference to a CONSORT diagram should be made.

[Answer 2c]

We included a reference to the CONSORT Diagram (Figure 2) (line 211+212)

[Comment 3]

CONSORT diagram: “Two or more cycles of FEC”. The meaning is not clear for interpretation. Sounds like 6 cycles also fit in this category and less than two are not.

[Answer 3]

Thank for this hint. That is a mistake by copy / paste. It should be said “More than three cycles FEC [n=10]” and we have corrected it (Figure 2)

[Comment 4a]

  1. 6 line 246 maximal age was 70 years and the inclusion criteria are less than 65. How many patients were included despite the inclusion criteria? Were there violations of other inclusion criteria?

[Answer 4a]

Thank for this hint. That is a mistake. It must be said “between 18 and 70 years of age were eligible”. We have corrected it in the text (2.2 Patient Population, line 105).

[Comment 4b]

p.6 line 246 percent of patients who were premenopausal do not match with percent in Table 1.

[Answer 4b]

Thank for this hint. It should be“41.4%”. Since we summarized peri- and premenopausal patients in one group. We have corrected it in the text (line 249).

[Comment 4c]

p.6 first paragraph. It will be more visually easy to perceive information if there will be a min-max range for tumor size and absolute numbers with percents for other values.

[Answer 4c]

The inclusion criteria prescribed 5 mm to 50 mm, and the actual range is 5 mm to 50 mm. Thus, we don’t see additional information by describing the range. In addition median tumor size of 1.9 cm implies that 50% of the patients had tumors >1.9 cm. We consider this as sufficient information about tumor size and distribution (see line 250).

[Comment 4d]

Table 1. HER2 status missing information is not consistent with the information from CONSORT diagram in which unavailable HER2 status was an exclusion criterion from per-protocol analysis.

[Answer 4d]

There were three patients, who fulfilled the two protocol violations “no HER2” and “insufficient chemotherapy”, simultaneously. Of course, these patients were counted only once. That explains the difference (18 cases in the CONSORT diagram, 21 cases without HER2-determination overall).

We have added the following words to the description of the CONSORT diagram to make it more clear: “Note, some patients had two (n=34) or three (n=4) protocol violations, simultaneously.” (line 245)

[Comment 4e]

Table 1. If you define Breast Surgery and Axillary surgery separately it would be appropriate to define Sentinel lymph node biopsy also.

[Answer 4e]

Actually, “mastectomy plus axillary dissection” was used as term to describe “modified radical mastectomy” in contrast to “breast conserving therapy”: we did not document numbers of axillary dissection and sentinel procedures. Thus, we think it would be appropriate to delete ALND and to refer to the type of breast surgery only.

We deleted “+ALND” from Table 1.

[Comment 4f]

p.8 line 257-258 numbers of patients with DFS events does not meet the numbers in table 1.

[Answer 4f]

Table 1 depicts every event that has been documented during the follow-up period, as DFS events are counted only those that were documented as first event as primary endpoint and, thus, included into the Kaplan-Meier-calculations.

[Comment 4g]

p.8 line 259 Seems like you mean disease free survival and not disease free interval.

[Answer 4g]

Disease free survival is correct (line 263).

[Comment 4h]

  1. 9 Toxicity paragraph. It is very hard to perceive information about adverse effects in absolute number of cycles performed. Maybe it would be better to present them in percentage of patients in whom particular adverse effect was developed.

[Answer 4h]

We understand your comment. However, we considered presenting the toxicities in relation to the patient numbers less realistic than presenting toxicities in relation to the chemotherapy courses administered.

Relevant toxicity per patient (leading to therapy discontinuation) can be derived from the fact that in the FEC-Doc arm 185 (14.3%), and in the FEC arm 98 (7.8%) patients, respectively, did not receive the entire chemotherapy (documented in CONSORT diagram, Figure 2).

Since evaluation of toxicity was not a primary endpoint but only of descriptive meaning, we would leave the table at is.

[Comment 5]

First sentence of conclusion states that this is the first phase 3 trial in NNBC which compared different risk assessment systems and that’s true, but the article null hypothesis was not about that statement and that’s why this sentence is irrelevant to the article and results

[Answer 5]

The comment is correct. However, that comparison was the intention of the trial and here we report on the first results of this trial. In order to make it more clear, in the second phrase we have added, that here we show first results.

[Major comments]

This is well designed study and the idea is clear and understandable. Nevertheless, from the start of the accrual till nowadays more than 20 years past. Nowadays we have modern standards of therapy with excellent oncologic outcomes which exceed numbers achieved in this study. In our daily practice we are using new surgical techniques and new regimens of systemic therapy which are obviously not taken into consideration as the trial was designed over 2 decades ago. It is an interesting work but it’ll be great if you will specify its meaning for routine clinical practice more precisely.

[Answer to Major comments]

Of course, this comment meets a critical point of the evaluation of this trial. However, we consider <7% recurrences after 5 years represent a rather low rate in this well-defined high risk cohort of node-negative breast cancer. It demonstrates, that patients with high risk of recurrence breast cancer can be treated effectively by using standard adjuvant chemotherapies. And, anthracycline-taxane sequences are still standard for most situations. We would expect a small additional benefit by using dose-dense EC (*4) followed by weekly paclitaxel (*12). As a clinically meaningful conclusion, we pointed out that according to our findings it is important to adhere to the planned treatment protocol avoiding unnecessary treatment discontinuations. In addition, we could have concluded that patients, who are not willing to tolerate the risk of taxane induced neurotoxicity; could choose an alternative taxane-free combination therapy (FEC*6) based on this data.

However, considering the short follow-up we would not over-interpret our data, therefore we were reluctant to discuss these issues in the paper.

To the editor:

The editors requested to add some additional references. We followed this request and added the following references:

  • Andre F, Ismaila N, Allison KH, Barlow WE, Collyar DE, Damodaran S, Henry NL, Jhaveri K, Kalinsky K, Kuderer NM, Litvak A, Mayer EL, Pusztai L, Raab R, Wolff AC, Stearns V. Biomarkers for Adjuvant Endocrine and Chemotherapy in Early-Stage Breast Cancer: ASCO Guideline Update. J Clin Oncol. 2022 Jun 1;40(16):1816-1837. doi: 10.1200/JCO.22.00069. Epub 2022 Apr 19. Erratum in: J Clin Oncol. 2022 Aug 1;40(22):2514. PMID: 35439025.

    Rationale: This are updated guidelines that still contain a recommendation with regard to uPA/PAI-1
  • Look MP, van Putten WL, Duffy MJ, Harbeck N, Christensen IJ, Thomssen C, Kates R, Spyratos F, Fernö M, Eppenberger-Castori S, Sweep CG, Ulm K, Peyrat JP, Martin PM, Magdelenat H, Brünner N, Duggan C, Lisboa BW, Bendahl PO, Quillien V, Daver A, Ricolleau G, Meijer-van Gelder ME, Manders P, Fiets WE, Blankenstein MA, Broët P, Romain S, Daxenbichler G, Windbichler G, Cufer T, Borstnar S, Kueng W, Beex LV, Klijn JG, O'Higgins N, Eppenberger U, Jänicke F, Schmitt M, Foekens JA. Pooled analysis of prognostic impact of urokinase-type plasminogen activator and its inhibitor PAI-1 in 8377 breast cancer patients. J Natl Cancer Inst. 2002 Jan 16;94(2):116-28. doi: 10.1093/jnci/94.2.116. PMID: 11792750.

    Rationale: see next
  • Look M, van Putten W, Duffy M, Harbeck N, Christensen IJ, Thomssen C, Kates R, Spyratos F, Fernö M, Eppenberger-Castori S, Fred Sweep CG, Ulm K, Peyrat JP, Martin PM, Magdelenat H, Brünner N, Duggan C, Lisboa BW, Bendahl PO, Quillien V, Daver A, Ricolleau G, Meijer-van Gelder M, Manders P, Edward Fiets W, Blankenstein M, Broët P, Romain S, Daxenbichler G, Windbichler G, Cufer T, Borstnar S, Kueng W, Beex L, Klijn J, O'Higgins N, Eppenberger U, Jänicke F, Schmitt M, Foekens J. Pooled analysis of prognostic impact of uPA and PAI-1 in breast cancer patients. Thromb Haemost. 2003 Sep;90(3):538-48. doi: 10.1160/TH-02-11-0264. Erratum in: Thromb Haemost. 2003 Nov;90(5):960. Bendah PO [corrected to Bendahl PO]. PMID: 12958624.

    Rationale: In this pooled analysis, data from 16 European groups were collected and evaluated , validating the original prospective data (Jänicke et al. 1991)
  • Bonneterre J, Roché H, Kerbrat P, Brémond A, Fumoleau P, Namer M, Goudier MJ, Schraub S, Fargeot P, Chapelle-Marcillac I. Epirubicin increases long-term survival in adjuvant chemotherapy of patients with poor-prognosis, node-positive, early breast cancer: 10-year follow-up results of the French Adjuvant Study Group 05 randomized trial. J Clin Oncol. 2005 Apr 20;23(12):2686-93. doi: 10.1200/JCO.2005.05.059. PMID: 15837983.

    Rationale: Reference of the standard regime that was used in the NNBC3-trial
  • Siegel RL, Miller KD, Wagle NS, Jemal A. Cancer statistics, 2023. CA Cancer J Clin. 2023 Jan;73(1):17-48. doi: 10.3322/caac.21763. PMID: 36633525.

    Rationale: The statement regarding excellent prognosis has to be underlined by a corresponding reference.
  • Sakr H, Hamed RH, Anter AH, Yossef T. Sequential docetaxel as adjuvant chemotherapy for node-positive or/and T3 or T4 breast cancer: clinical outcome (Mansoura University). Med Oncol. 2013 Mar;30(1):457. doi: 10.1007/s12032-013-0457-3. Epub 2013 Jan 16. PMID: 23322524.

    Rationale: This study provides additional prospective data to the comparison FEC-Doc vs FEC, here demonstrating the benefit of docetaxel in patients with advanced breast cancer. Similarly as in our study, the benefit cannot be seen in the early follow-up.
  • Kerbrat P, Desmoulins I, Roca L, Levy C, Lortholary A, Marre A, Delva R, Rios M, Viens P, Brain É, Serin D, Edel M, Debled M, Campone M, Mourret-Reynier MA, Bachelot T, Foucher-Goudier MJ, Asselain B, Lemonnier J, Martin AL, Roché H. Optimal duration of adjuvant chemotherapy for high-risk node-negative (N-) breast cancer patients: 6-year results of the prospective randomised multicentre phase III UNICANCER-PACS 05 trial (UCBG-0106). Eur J Cancer. 2017 Jul;79:166-175. doi: 10.1016/j.ejca.2017.03.004. Epub 2017 May 11. PMID: 28501763.

    Rationale: Relation of our observation to known data. We addd the phrase” However, this result disagrees with data from the PACS05 trial that did not show a difference in efficacy between 4 and 6 courses FEC in a similar setting”
  • Jackisch C, Cortazar P, Geyer CE Jr, Gianni L, Gligorov J, Machackova Z, Perez EA, Schneeweiss A, Tolaney SM, Untch M, Wardley A, Piccart M. Risk-based decision-making in the treatment of HER2-positive early breast cancer: Recommendations based on the current state of knowledge. Cancer Treat Rev. 2021 Sep;99:102229. doi: 10.1016/j.ctrv.2021.102229. Epub 2021 May 20. PMID: 34139476.

    Rationale: The authors demonstrate the differentiated and successful treatment approaches in HER2-positive early breast cancer.
  • Petrelli F, Bertaglia V, Parati MC, Borgonovo K, De Silva P, Luciani A, Novello S, Scartozzi M, Emens LA, Solinas C. Adjuvant chemotherapy for resected triple negative breast cancer patients: A network meta-analysis. Breast. 2023 Feb;67:8-13. doi: 10.1016/j.breast.2022.12.004. Epub 2022 Dec 15. PMID: 36549170; PMCID: PMC9792383.

    Rationale: This data provides information on effects of adjuvant chemotherapy in early triple-negative breast cancer.

Reviewer 2 Report

UPA and PAI-1 are among the best validated prognostic biomarkers in breast cancer so including them in therapeutic decision is essential. 

The study has a excellent design (multicenter, prospective, randomized, not blinded, controlled  trial) and the conclusions are well sustained by the results. 

However it is not clear how classifying breast tumors according uPA and PAI-1 is superior or inferior to gene expression signatures tests. Maybe one possible benefit is using PAI-1 and uPA as supplementary markers for nodal status as well. 

It would also be interesting to discuss the secondary effects of FEC-Doc  in comparison with the new chemotherapy regimens. 

Author Response

The reviewers' comments are answered point by point and the responses are indicated in blue. The changes made are referenced to page and paragraph from the resubmitted manuscript

The changes are simultaneously applied and indicated in the manuscript's new version according to the Journal´s guidelines.

Reviewer 2:

[Comment 1]

UPA and PAI-1 are among the best validated prognostic biomarkers in breast cancer so including them in therapeutic decision is essential. 

[Answer 1]: That was the basis of the original idea of this trial.

[Comment 2]

The study has an excellent design (multicenter, prospective, randomized, not blinded, controlled trial) and the conclusions are well sustained by the results. 

[Answer 2]: We agree. Thank you.

[Comment 3]

However it is not clear how classifying breast tumors according uPA and PAI-1 is superior or inferior to gene expression signatures tests. Maybe one possible benefit is using PAI-1 and uPA as supplementary markers for nodal status as well. 

[Answer 3]

Although that would have been really an interesting question, it was not the question of the trial since gene expression profile were developed only during and after the NNBC-3-trial. We have stored a relevant proportion of the tumour specimen such that we will be able to perform such a study in future.

There is data upon the prognostic value of uPA/PAI-1 in node-positive disease (Jänicke et al., 1991), however, we were not courageous enough to include node-positive patients into NNBC-3 trial as it has been done e.g. in the MINDACT trial.

[Comment 4]

It would also be interesting to discuss the secondary effects of FEC-Doc in comparison with the new chemotherapy regimens.

[Answer 4]

We discussed the FEC-Doc regimen in the context of similar trials including the Oxford overview to the effect of taxanes (line 348+) and we added some other relevant publications (PACS-05 etc., line 339+). We think a more extensive discussion would only be relevant if we have reached a longer follow-up with more robust data.

To the editors:

The editors requested to add some additional references. We followed this request and added the following references:

  • Andre F, Ismaila N, Allison KH, Barlow WE, Collyar DE, Damodaran S, Henry NL, Jhaveri K, Kalinsky K, Kuderer NM, Litvak A, Mayer EL, Pusztai L, Raab R, Wolff AC, Stearns V. Biomarkers for Adjuvant Endocrine and Chemotherapy in Early-Stage Breast Cancer: ASCO Guideline Update. J Clin Oncol. 2022 Jun 1;40(16):1816-1837. doi: 10.1200/JCO.22.00069. Epub 2022 Apr 19. Erratum in: J Clin Oncol. 2022 Aug 1;40(22):2514. PMID: 35439025.

    Rationale: This are updated guidelines that still contain a recommendation with regard to uPA/PAI-1
  • Look MP, van Putten WL, Duffy MJ, Harbeck N, Christensen IJ, Thomssen C, Kates R, Spyratos F, Fernö M, Eppenberger-Castori S, Sweep CG, Ulm K, Peyrat JP, Martin PM, Magdelenat H, Brünner N, Duggan C, Lisboa BW, Bendahl PO, Quillien V, Daver A, Ricolleau G, Meijer-van Gelder ME, Manders P, Fiets WE, Blankenstein MA, Broët P, Romain S, Daxenbichler G, Windbichler G, Cufer T, Borstnar S, Kueng W, Beex LV, Klijn JG, O'Higgins N, Eppenberger U, Jänicke F, Schmitt M, Foekens JA. Pooled analysis of prognostic impact of urokinase-type plasminogen activator and its inhibitor PAI-1 in 8377 breast cancer patients. J Natl Cancer Inst. 2002 Jan 16;94(2):116-28. doi: 10.1093/jnci/94.2.116. PMID: 11792750.

    Rationale: see next
  • Look M, van Putten W, Duffy M, Harbeck N, Christensen IJ, Thomssen C, Kates R, Spyratos F, Fernö M, Eppenberger-Castori S, Fred Sweep CG, Ulm K, Peyrat JP, Martin PM, Magdelenat H, Brünner N, Duggan C, Lisboa BW, Bendahl PO, Quillien V, Daver A, Ricolleau G, Meijer-van Gelder M, Manders P, Edward Fiets W, Blankenstein M, Broët P, Romain S, Daxenbichler G, Windbichler G, Cufer T, Borstnar S, Kueng W, Beex L, Klijn J, O'Higgins N, Eppenberger U, Jänicke F, Schmitt M, Foekens J. Pooled analysis of prognostic impact of uPA and PAI-1 in breast cancer patients. Thromb Haemost. 2003 Sep;90(3):538-48. doi: 10.1160/TH-02-11-0264. Erratum in: Thromb Haemost. 2003 Nov;90(5):960. Bendah PO [corrected to Bendahl PO]. PMID: 12958624.

    Rationale: In this pooled analysis, data from 16 European groups were collected and evaluated , validating the original prospective data (Jänicke et al. 1991)
  • Bonneterre J, Roché H, Kerbrat P, Brémond A, Fumoleau P, Namer M, Goudier MJ, Schraub S, Fargeot P, Chapelle-Marcillac I. Epirubicin increases long-term survival in adjuvant chemotherapy of patients with poor-prognosis, node-positive, early breast cancer: 10-year follow-up results of the French Adjuvant Study Group 05 randomized trial. J Clin Oncol. 2005 Apr 20;23(12):2686-93. doi: 10.1200/JCO.2005.05.059. PMID: 15837983.

    Rationale: Reference of the standard regime that was used in the NNBC3-trial
  • Siegel RL, Miller KD, Wagle NS, Jemal A. Cancer statistics, 2023. CA Cancer J Clin. 2023 Jan;73(1):17-48. doi: 10.3322/caac.21763. PMID: 36633525.

    Rationale: The statement regarding excellent prognosis has to be underlined by a corresponding reference.
  • Sakr H, Hamed RH, Anter AH, Yossef T. Sequential docetaxel as adjuvant chemotherapy for node-positive or/and T3 or T4 breast cancer: clinical outcome (Mansoura University). Med Oncol. 2013 Mar;30(1):457. doi: 10.1007/s12032-013-0457-3. Epub 2013 Jan 16. PMID: 23322524.

    Rationale: This study provides additional prospective data to the comparison FEC-Doc vs FEC, here demonstrating the benefit of docetaxel in patients with advanced breast cancer. Similarly as in our study, the benefit cannot be seen in the early follow-up.
  • Kerbrat P, Desmoulins I, Roca L, Levy C, Lortholary A, Marre A, Delva R, Rios M, Viens P, Brain É, Serin D, Edel M, Debled M, Campone M, Mourret-Reynier MA, Bachelot T, Foucher-Goudier MJ, Asselain B, Lemonnier J, Martin AL, Roché H. Optimal duration of adjuvant chemotherapy for high-risk node-negative (N-) breast cancer patients: 6-year results of the prospective randomised multicentre phase III UNICANCER-PACS 05 trial (UCBG-0106). Eur J Cancer. 2017 Jul;79:166-175. doi: 10.1016/j.ejca.2017.03.004. Epub 2017 May 11. PMID: 28501763.

    Rationale: Relation of our observation to known data. We addd the phrase” However, this result disagrees with data from the PACS05 trial that did not show a difference in efficacy between 4 and 6 courses FEC in a similar setting”
  • Jackisch C, Cortazar P, Geyer CE Jr, Gianni L, Gligorov J, Machackova Z, Perez EA, Schneeweiss A, Tolaney SM, Untch M, Wardley A, Piccart M. Risk-based decision-making in the treatment of HER2-positive early breast cancer: Recommendations based on the current state of knowledge. Cancer Treat Rev. 2021 Sep;99:102229. doi: 10.1016/j.ctrv.2021.102229. Epub 2021 May 20. PMID: 34139476.

    Rationale: The authors demonstrate the differentiated and successful treatment approaches in HER2-positive early breast cancer.
  • Petrelli F, Bertaglia V, Parati MC, Borgonovo K, De Silva P, Luciani A, Novello S, Scartozzi M, Emens LA, Solinas C. Adjuvant chemotherapy for resected triple negative breast cancer patients: A network meta-analysis. Breast. 2023 Feb;67:8-13. doi: 10.1016/j.breast.2022.12.004. Epub 2022 Dec 15. PMID: 36549170; PMCID: PMC9792383.

    Rationale: This data provides information on effects of adjuvant chemotherapy in early triple-negative breast cancer.

Reviewer 3 Report

This manuscript presents the results of a randomized, controlled phase III trial comparing 6 cycles of FEC with 3 cycles of FEC followed by 3 cycles of docetaxel (DTX) in patients with lymph node-negative breast cancer. Unfortunately, the trial showed no survival benefit with the combination of DTX and FEC compared to FEC alone. Several reasons for the negative results in this trial were discussed. My comments are as follows.

1.       This study is well designed, organized, and analyzed. Previous studies have shown that the combination of DTX and anthracyclines improves survival in node-negative breast cancer patients; the main reason for the lack of survival benefit with DTX is that discontinuation of DTX after FEC is approximately twice that of FEC alone?

2.       There are several possible reasons for the negative results with DTX: 5-year DFS and OS were >90% for both FEC alone and with DTX, but there are still cases of recurrence with FEC alone or with DTX after FEC.

3.       In this study, there was no survival benefit from DTX followed by FEC. However, the key question is how to prevent recurrence in postoperative adjuvant therapy for node-negative patients, a high-risk group. What is the strategy for these patients? These points also need to be discussed.

4.       Are these recurrent patients characterized in any way by tumor subtype?

5.       Misspelled word (L 189).

Author Response

The reviewers' comments are answered point by point and the responses are indicated in blue. The changes made are referenced to page and paragraph from the resubmitted manuscript.

The changes are simultaneously applied and indicated in red in the manuscript's new version according to the Journal´s guidelines.

Reviewer 3:

This manuscript presents the results of a randomized, controlled phase III trial comparing 6 cycles of FEC with 3 cycles of FEC followed by 3 cycles of docetaxel (DTX) in patients with lymph node-negative breast cancer. Unfortunately, the trial showed no survival benefit with the combination of DTX and FEC compared to FEC alone. Several reasons for the negative results in this trial were discussed. My comments are as follows.

[Comment 1]

This study is well designed, organized, and analyzed. Previous studies have shown that the combination of DTX and anthracyclines improves survival in node-negative breast cancer patients; the main reason for the lack of survival benefit with DTX is that discontinuation of DTX after FEC is approximately twice that of FEC alone?

[Answer 1]

This comment is correct. Actually we discussed this point in line 347-351.

[Comment 2]

There are several possible reasons for the negative results with DTX: 5-year DFS and OS were >90% for both FEC alone and with DTX, but there are still cases of recurrence with FEC alone or with DTX after FEC.

[Answer 2]

Basically correct. However, by the multivariate analyses we showed that particularly insufficient administration of chemotherapy is the strongest prognostic factor for recurrence. That underlines that we identified the right patients for chemotherapy independent from the type of treatment.

[Comment 3]

In this study, there was no survival benefit from DTX followed by FEC. However, the key question is how to prevent recurrence in postoperative adjuvant therapy for node-negative patients, a high-risk group. What is the strategy for these patients? These points also need to be discussed.

[Answer 3]

That is indeed an interesting point. Therefore we added a paragraph:

High levels of uPA/PAI-1 are predictive for an enhanced benefit from adjuvant chemotherapy [Harbeck et al., 2002]. However, despite treating with effective therapies we still observe some recurrences. Incompletely delivered chemotherapy has been demonstrated to be the most important predictor; however, multiple other reasons can be discussed like primary resistance to chemotherapy, failing diagnosis of metastatic disease, or suboptimal design or lack of adherence for endocrine therapy. In order to further reduce the risk of recurrence, today more effective therapies including targeted therapies would be administered. (line 357-363)

 [Comment 4]

Are these recurrent patients characterized in any way by tumor subtype?

[Answer 4]

We did an additional subgroup analysis of luminal cancers only and found similar results to the entire cohort. (suppl. Figure S3)

[Comment 5]

Misspelled word (L 189).

[Answer 5]

We did not find the misspelling, please specify.

[Comment 6]

Are these recurrent patients characterized in any way by tumor subtype?

[Answer 6]

We looked for that and did not find any differential effect (see Figure 4).

To the editor:

The editors requested to add some additional references. We followed this request and added the following references:

  • Andre F, Ismaila N, Allison KH, Barlow WE, Collyar DE, Damodaran S, Henry NL, Jhaveri K, Kalinsky K, Kuderer NM, Litvak A, Mayer EL, Pusztai L, Raab R, Wolff AC, Stearns V. Biomarkers for Adjuvant Endocrine and Chemotherapy in Early-Stage Breast Cancer: ASCO Guideline Update. J Clin Oncol. 2022 Jun 1;40(16):1816-1837. doi: 10.1200/JCO.22.00069. Epub 2022 Apr 19. Erratum in: J Clin Oncol. 2022 Aug 1;40(22):2514. PMID: 35439025.

    Rationale: This are updated guidelines that still contain a recommendation with regard to uPA/PAI-1
  • Look MP, van Putten WL, Duffy MJ, Harbeck N, Christensen IJ, Thomssen C, Kates R, Spyratos F, Fernö M, Eppenberger-Castori S, Sweep CG, Ulm K, Peyrat JP, Martin PM, Magdelenat H, Brünner N, Duggan C, Lisboa BW, Bendahl PO, Quillien V, Daver A, Ricolleau G, Meijer-van Gelder ME, Manders P, Fiets WE, Blankenstein MA, Broët P, Romain S, Daxenbichler G, Windbichler G, Cufer T, Borstnar S, Kueng W, Beex LV, Klijn JG, O'Higgins N, Eppenberger U, Jänicke F, Schmitt M, Foekens JA. Pooled analysis of prognostic impact of urokinase-type plasminogen activator and its inhibitor PAI-1 in 8377 breast cancer patients. J Natl Cancer Inst. 2002 Jan 16;94(2):116-28. doi: 10.1093/jnci/94.2.116. PMID: 11792750.

    Rationale: see next
  • Look M, van Putten W, Duffy M, Harbeck N, Christensen IJ, Thomssen C, Kates R, Spyratos F, Fernö M, Eppenberger-Castori S, Fred Sweep CG, Ulm K, Peyrat JP, Martin PM, Magdelenat H, Brünner N, Duggan C, Lisboa BW, Bendahl PO, Quillien V, Daver A, Ricolleau G, Meijer-van Gelder M, Manders P, Edward Fiets W, Blankenstein M, Broët P, Romain S, Daxenbichler G, Windbichler G, Cufer T, Borstnar S, Kueng W, Beex L, Klijn J, O'Higgins N, Eppenberger U, Jänicke F, Schmitt M, Foekens J. Pooled analysis of prognostic impact of uPA and PAI-1 in breast cancer patients. Thromb Haemost. 2003 Sep;90(3):538-48. doi: 10.1160/TH-02-11-0264. Erratum in: Thromb Haemost. 2003 Nov;90(5):960. Bendah PO [corrected to Bendahl PO]. PMID: 12958624.

    Rationale: In this pooled analysis, data from 16 European groups were collected and evaluated , validating the original prospective data (Jänicke et al. 1991)
  • Bonneterre J, Roché H, Kerbrat P, Brémond A, Fumoleau P, Namer M, Goudier MJ, Schraub S, Fargeot P, Chapelle-Marcillac I. Epirubicin increases long-term survival in adjuvant chemotherapy of patients with poor-prognosis, node-positive, early breast cancer: 10-year follow-up results of the French Adjuvant Study Group 05 randomized trial. J Clin Oncol. 2005 Apr 20;23(12):2686-93. doi: 10.1200/JCO.2005.05.059. PMID: 15837983.

    Rationale: Reference of the standard regime that was used in the NNBC3-trial
  • Siegel RL, Miller KD, Wagle NS, Jemal A. Cancer statistics, 2023. CA Cancer J Clin. 2023 Jan;73(1):17-48. doi: 10.3322/caac.21763. PMID: 36633525.

    Rationale: The statement regarding excellent prognosis has to be underlined by a corresponding reference.
  • Sakr H, Hamed RH, Anter AH, Yossef T. Sequential docetaxel as adjuvant chemotherapy for node-positive or/and T3 or T4 breast cancer: clinical outcome (Mansoura University). Med Oncol. 2013 Mar;30(1):457. doi: 10.1007/s12032-013-0457-3. Epub 2013 Jan 16. PMID: 23322524.

    Rationale: This study provides additional prospective data to the comparison FEC-Doc vs FEC, here demonstrating the benefit of docetaxel in patients with advanced breast cancer. Similarly as in our study, the benefit cannot be seen in the early follow-up.
  • Kerbrat P, Desmoulins I, Roca L, Levy C, Lortholary A, Marre A, Delva R, Rios M, Viens P, Brain É, Serin D, Edel M, Debled M, Campone M, Mourret-Reynier MA, Bachelot T, Foucher-Goudier MJ, Asselain B, Lemonnier J, Martin AL, Roché H. Optimal duration of adjuvant chemotherapy for high-risk node-negative (N-) breast cancer patients: 6-year results of the prospective randomised multicentre phase III UNICANCER-PACS 05 trial (UCBG-0106). Eur J Cancer. 2017 Jul;79:166-175. doi: 10.1016/j.ejca.2017.03.004. Epub 2017 May 11. PMID: 28501763.

    Rationale: Relation of our observation to known data. We addd the phrase” However, this result disagrees with data from the PACS05 trial that did not show a difference in efficacy between 4 and 6 courses FEC in a similar setting”
  • Jackisch C, Cortazar P, Geyer CE Jr, Gianni L, Gligorov J, Machackova Z, Perez EA, Schneeweiss A, Tolaney SM, Untch M, Wardley A, Piccart M. Risk-based decision-making in the treatment of HER2-positive early breast cancer: Recommendations based on the current state of knowledge. Cancer Treat Rev. 2021 Sep;99:102229. doi: 10.1016/j.ctrv.2021.102229. Epub 2021 May 20. PMID: 34139476.

    Rationale: The authors demonstrate the differentiated and successful treatment approaches in HER2-positive early breast cancer.
  • Petrelli F, Bertaglia V, Parati MC, Borgonovo K, De Silva P, Luciani A, Novello S, Scartozzi M, Emens LA, Solinas C. Adjuvant chemotherapy for resected triple negative breast cancer patients: A network meta-analysis. Breast. 2023 Feb;67:8-13. doi: 10.1016/j.breast.2022.12.004. Epub 2022 Dec 15. PMID: 36549170; PMCID: PMC9792383.

    Rationale: This data provides information on effects of adjuvant chemotherapy in early triple-negative breast cancer.
